# Marine Litter on the Coast of the Algarve: Main Sources and Distribution Using a Modeling Approach

Eloah Rosas [1,*], Flávio Martins [1,2] and João Janeiro [1]

1   Centro de Investigação Marinha e Ambiental (CIMA), Campus de Gambelas, University of Algarve, 8005-139 Faro, Portugal; fmartins@ualg.pt (F.M.); jmjaneiro@ualg.pt (J.J.)
2   Instituto Superior de Engenharia (ISE), Campus de Gambelas, University of Algarve, 8005-139 Faro, Portugal
*   Correspondence: egrosas@ualg.pt

**Abstract:** The accumulation of floating marine litter poses a serious threat to the global environment and the economy all over the world, particularly of coastal municipalities that rely on tourism and recreational activities. Data of marine litter is thus crucial, but is usually limited, and can be complemented with modelling results. In this study, the operational modelling system of Algarve (SOMA) was combined with a Lagrangian particle-tracking model and blended with scarce litter monitoring data, to provide first insights into the distribution and accumulation of floating marine litter on the Algarve coast. Different meteo-oceanographic conditions, sources regions and wind drift behaviors were considered. Field data and model results show a considerable concentration of marine litter along the beaches and coastal regions. The model also suggests that oceanographic conditions and wind drift have a great influence on the transport and accumulation rate of the floating marine litter on the coast, with the highest rates of accumulation during the winter and the counter current period, concentrated mostly on the south-western coast of the Algarve.

**Keywords:** Algarve; floating marine litter; numerical models; SOMA





## 1. Introduction

Litter in marine environment has become a major worldwide concern. Anthropogenic litter enters into the marine environment by way of various land-based (e.g., holidaymakers and beachgoers, which can cause littering directly on the coast, and riverine litter) and at-sea sources (e.g., shipping and fishing activities and aquaculture installation) [1,2]. After entering the ocean, marine litter items remain on the sea surface and are then transported by oceanic and atmospheric dynamics until they sink to the seafloor. They can also get deposited on the shore or degrade over time. This research will only focus on floating marine litter along the coastal water surface. Marine litter floating in the open seas and coastal waters pose a great risk to the marine environment. Additionally, their presence on the shores can be a challenging phenomenon for coastal municipalities that rely on tourism and recreational activities [3]. Litter larger than 2.5 cm, known as macro-litter, reduces the appeal of beach destinations and consequently reduces tourism and forces local authorities to invest in beach cleaning strategies [4].

Over the last decade, major global efforts have been made to better address problematic marine litter; like, for example, the Honolulu Convention, United Nations Environment Program Regional Sea (UNEP) and the Global Partnership on Marine Debris (GPML). In Europe, marine litter is acknowledged by the Marine Strategy Framework Directive (MSFD). The MSFD is a framework which requires the members of a state to develop and apply programmes of measures to achieve or maintain a Good Environmental Status (GES) in European Seas [5]. Moreover, the MSFD GES technical subgroup on marine litter identified various priority research for a better assessment of marine litter, such as: evaluating the factors that affect the transport and fate, development of standardization monitoring methodologies, and use of modeling to assist in monitoring and management.

In this context, monitoring programs should assess litter found on shorelines, floating on the sea surface, or deposited on the sea floor, as well as analyzing it in terms of spatial distributions, accumulation, and litter sources and types [6]. Ocean circulation models coupled with Lagrangian particle-tracking models have proved to be valuable tool for identifying trajectories, accumulation regions of marine litter in the oceans [7,8], regional seas [3,9], and coastal regions [4,10]. Among these models, typically Lagrangian particle tracking schemes are used, where the marine litter is represented by numerical particles. These particles are transported due to winds, waves and currents, and are diffused by turbulence [11].

The Portuguese coast is exposed to marine litter accumulation from diverse land and marine sources, due to the population concentration (more than half of the population) living near to the coastline, intense tourism, high fishing activities, and it also being an important shipping route [12]. Some scientific surveys have been carried out to quantify and characterize marine litter on the Portuguese coast [13,14]. Only recently has ocean modelling been applied to investigate the trajectory of floating litter, and only in the northern Iberian waters [15]. The study observes that the distribution and fate of floating litter is influenced by seasonal circulation. Nevertheless, despite all the efforts, information on the distribution of marine litter along the Portuguese coast remain scarce, and principally located in the southern region. As the south coast of Portugal is the most touristic region, a deeper understanding of the presence and fate of marine litter in this region is crucial.

In this context, this research aims to give a step forward in the comprehension of the floating marine macro litter on the Algarve coast by applying a numerical model and combining the results with in-situ measurements. First, local meteo-oceanographic conditions, hindcasted by the Algarve Operational Modelling and Monitoring System (SOMA), were used to evaluate the Lagrangian transport of floating marine macro litter which has originated from different sources (land- and sea-based), including wind drift and beaching effects. From these simulations, potential trajectories and hotspots for these pollutants could be assessed. Second, in-situ beach litter measurements collected following standardized approaches were analyzed using the Litter Analyst software, to provide reliable and valuable information to qualitatively evaluate and complement the model results.

This article is structured in five Sections: 1. Introduction, the present section; 2. Materials and Methods with a brief description of the study area, the model used as well as the set-up of the model and the methodology used in the collection and statistical analysis of beach monitoring; 3. Results, including results of the modelling activities and the in-situ surveys; 4. Discussion; and 5. Conclusions.

## 2. Materials and Methods

### 2.1. Study Area

The Algarve coast, located at the south-western Iberian Peninsula, presents complex oceanographic conditions and morphological features. In this region the ocean circulation has a marked seasonal pattern [16]. In winter, the ocean circulation is influenced by southerly winds, with a presence of poleward surface circulation along the west coast. During the late spring–summer, the prevailing northerly winds induce upwelling conditions. This period is characterized by a cold southward flow along the western coast, whereas westerly winds drive off the upwelled water eastward along the southern coast [17]. However, under periods of northerly wind relaxation, a coastal counter-current occurs in the southern coast, carrying warm waters westward, and eventually turning northward at Cape St Vincent (CSV) [18]. In [16], a review of the main oceanographic features of the region is presented.

The Algarve coast is a region with important ecological interest, encompassing important natural parks with high biodiversity, such as the Ria Formosa Natural Park (PNRF) and the Southwest Alentejo and Vicentine Coast Natural Park (PNSACV). Its economy and regional development is based on the blue economy, depending mainly on tourism

(in particular "sun and sand") and fishery activities [18]. It is important to highlight that around 4.7 million tourists visit this coastal region every year [19]. Diverse studies have reported the presence of marine litter on the sea floor [20] and beaches [21], as well as in the stomachs of aquatic birds [22].

### 2.2. Modeling of Floating Marine Litter
### 2.2.1. Hydrodynamic Model

The Algarve Operational Modelling and Monitoring System (SOMA) was used to simulate the hydrodynamic conditions of the domain show in Figure 1. SOMA is the operational regional downscales model for SW Iberia, maintained by the Marine and Environmental Research Centre (CIMA) of the University of Algarve and is in its second year of operation, producing 3 days of daily forecasts. SOMA is based on the MOHID (derived from the Portuguese abbreviation of "MODelo HIDrodinâmico") water modelling system, a three-dimensional hydrodynamic modelling system that integrates diverse modules in an object-oriented programming approach. This approach isolates different parts of the code and allows communication among them using simple and robust interfaces [23]. The hydrodynamic module computes and updates the flow information solving the primitive Navier–Stokes equations in the three-dimensional space for incompressible fluids. The Navier–Stokes equations is applied to describe the motion of fluids, for example in ocean modelling, approximations are used to solve these equations, written in the coordinate frame of reference fixed to the rotating earth, which, together with an equation of state (relating pressure, temperature, and salinity) and equations for turbulent quantities, allow us to compute the ocean velocity fields.

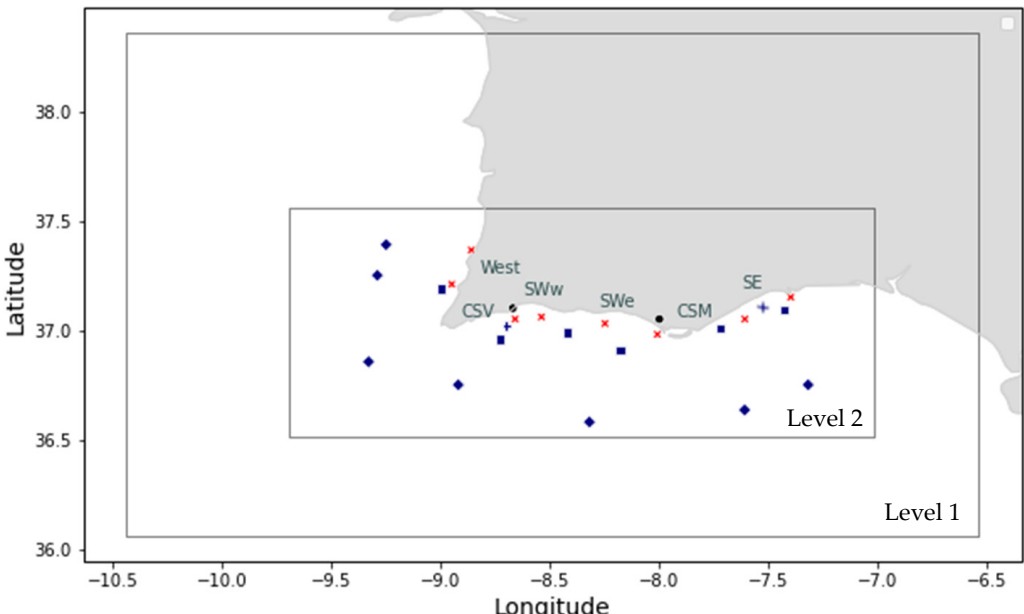

**Figure 1.** The two domains that composed SOMA (Level 1 and Level 2), and floating litter emission points used in this work: touristic beaches and rivers (red x), fishing grounds (blue squares), aquaculture sites (blue crosses), major shipping lanes (blue diamond), and sampling data (black circle).

SOMA is composed of two nested domains. The first domain, hereby called Level 1 (Figure 1), spans from 36° to 38.36° N and 6.54° to 10.44° W, with a horizontal resolution of 3 km and a time step of 30 s. The main domain (Level 2) covers the entire Algarve region with 1 km spatial resolution and a time step of 15 s. Both levels share the same vertical resolution with 50 vertical levels with a variable vertical resolution decreasing from 1 m at the surface down to 20 m at the bottom. More information on SOMA and its validation can be found on [23].

In this study, SOMA downscales the product of Iberia Biscay Ireland Regional Operational Oceanographic System (IBI-ROOS), which provides a daily forecast of water current velocities at 3 km horizontal resolution. The atmosphere used is the Regional Weather Forecasting System (SKIRON) at a horizontal resolution of 5 km.

To simulate the floating marine litter trajectories, the Lagrangian module based on the MOHID Water system is used. The Lagrangian framework describes the motion of virtual particles in the velocity field as a function of time [24]. The spatial transport of the particles is the sum of four velocities:

$$\frac{dx_i}{dt} = U_i(x_i, t) \tag{1}$$

where $x_i$ denotes the particle position at time t and:

$$U_i = u_{1i} + C_D u_{2i} + u_{3i} + u_{4i} \tag{2}$$

Looking at Equation (2), $u_{1i}$ is the surface current velocity from the hydrodynamic module. In this study, both hydrodynamic and Lagrangian modules run simultaneously and there is an exchange of turbulence information in real time between both models, thus increasing the accuracy of the computed trajectories [25]. In addition, the multi-mesh approach ensures that the high-resolution hydrodynamics (Level 2) is selected whenever the particles move into their geographical boundaries. $u_{2i}$ is the wind velocity at a height of 10 m over the sea surface, and $C_D$ is the user-defined wind drag coefficient. The wind drag factor mostly compensates for the lack of wind stress in the upper few centimeters of the hydrodynamic models to drive the marine litter immersed in the water. This approach is applied in hydrodynamic models with the vertical thickness of the top layer being not sufficiently small. This situation would require a resolution at the surface of the order of a few centimeters, which is currently not computationally feasible. This drag velocity is also useful to account for the effect of the wind in the exposed part of emerged litter. Many studies reported that the wind-induced speed of floating marine litter at the sea surface, and values of wind drag coefficient, normally varies from 1% to 5% [3,4,10]. $u_{3i}$ is the random velocity due to the diffusion transport, and finally, $u_{4i}$ is the stokes drift. Although the stokes drift has been reported as an important mechanism that affects the trajectories of floating marine litter, in this work it was not considered because up to now, no regional product was available to supply wave data to the SOMA system.

Beaching of particles was considered, hence the change of litter particle status from "on surface" to "beached", and was decided by checking in each time-step whether the particle crosses a band within 5 m close of the coastline. This 5 m value was decided based on the average beach profile in the region. Furthermore, the beaching status of a particle may not be permanent. Beached particles do not move but have a chance to be washed back to the coastal water at each time step. In this work, this probability was set at 50%, and this value is based on a pragmatic choice, since limited information of beaching of marine litter is available. In addition, sinking and degradation of particles were not incorporated in this work, as a result of the necessity of improved studies for further modelling.

### 2.2.2. Model Set Up

Floating marine macro litter is considered in the model from marine- and land-based sources (Figure 1). For land-based sources, particles were emitted in the water close to the 4 cities with higher urbanization and tourism activities: Lagos, Albufeira, Faro, and Tavira [26]. On the west coast, although there is lower urban pressure, two emission points were added to ascertain the presence of litter in the southwest: Alentejo and Vicentina Coast Natural Park. Emission points at the Guadiana and Arade estuaries were also considered, since they serve large urban centres and present high anthropogenic pressure. The former borders several cities in Portugal and Spain, while the latter borders Portimão, one of the largest cities in the Algarve.

The activities of maritime transport, fishing, and aquaculture were also considered as sources of marine litter. According to data from the Automatic Identification System (AIS) in [27], the highest density of maritime traffic in the Iberian southwest is observed on the main commercial navigation route that connects the Mediterranean to the North Atlantic waters, while in coastal waters moderate maritime traffic is related to inshore fishing. In this context, seven emission points were positioned along the main offshore corridor of the study area, while at coastal water, 6 points were added in fishing grounds with importance for the local and commercial fleets observed by [28]. Two more emission points were placed close to the aquaculture sites, shown on the DGRM (Directorate-General for Natural Resources, Safety and Marine Service) geoportal [29].

To understand the role of meteo-oceanographic conditions in the trajectories and accumulation of floating marine litter, three scenarios were simulated, and all simulations ran for 15 days. The simulations commenced on the 1 January, 15 September and 7 October 2017, and are referred to here as scenarios 1, 2, and 3, respectively. These periods were chosen through previous analysis of satellite images of the sea surface temperature, in which it was possible to observe different surface circulation patterns according to the description in Section 2. These scenarios represent typical conditions of winter circulation and summer circulation with upwelling and counter currents, respectively.

In each location, 1000 particles were emitted instantaneously at the sea surface, where they remained for the whole run period. For each scenario, two simulations were performed; one considering only the effect of wind on the hydrodynamic results, and another with an additional wind drift effect applied to the particles, as explained in the previous section. For this purpose, 3% of the wind speed was added to the particles as recommended by [4,30]. The emission properties are summarized in Table 1.

**Table 1.** Parameters and values used in the simulations.

| Parameter | Values |
| --- | --- |
| Wind Drift coefficient | 0 and 3% |
| Beaching | 5 m close to the shoreline |
| Source location | 23 points of emission |
| Number of particles | 1000 particles in each point of emission |

### 2.3. Beach Litter Monitoring

Along the Algarve coast, suitable field data of floating marine litter to validate the simulations were limited. In this study, the surveyed beaches were selected based on data availability to qualitatively assess the model's accumulation accuracy. Firstly, Directorate-General for Natural Resources, Safety and Marine Service (DGRM), and the Portuguese Environment Agency (APA) have been monitoring litter on two Algarve beaches since 2013; Faro Island and Praia da Batata beach, according with the framework of the Convention for the Protection of the Marine Environment of the North-East Atlantic (known as OSPAR Convention). The data composed of the amount of litter items documented for each of the 112 predefined litter types throughout surveys of 100 m of shoreline, categorized by material (e.g., plastic, paper, metal, glass) and sources (e.g., tourism, fishing, shipping, other) [30]. The 100 m surveys are done at fixed positions and 4 times a year [30]. To qualitatively validate the model, data collected in winter, summer, and autumn surveys of 2017 were analysed by the Litter Analyst software and were used to compare with the beaching results for all three simulations.

APA also provided field data collected between 1 January 2013 to 31 December 2017 on both beaches: Faro Island and Praia da Batata. This study took the opportunity to use the Litter Analyst tool to also assess the long-term field data to complement the model results and gain a better insight into the litter pollution along the Algarve coastlines.

Beach Litter Data Analysis

The statistical analysis of beach data was performed by the new tool Litter Analyst (version 3.0). This software was created with the aim of supporting researchers and policy makers to analyse and monitor litter along beaches. For a full description of the software, see [31].

For this study, the descriptive statistics—median, arithmetic mean, and the relative abundance of each item (%)—relate to total litter items and were computed for the total amount, types, and sources of litter recorded on Faro Island and Praia da Batata between 2013 to 2017. Moreover, the similar descriptive statistics were also computed for the most common litter types found on the shores, which together contribute to 80% of the total litter items recorded on a beach during a certain period (called here the top-80% litter). Trend analyses were also calculated using Theil-Sen slope estimations and Mann–Kendall significance tests for each survey site. Statistical significance was established at a probability level ($\alpha$) of 0.05.

## 3. Results

To assess the trajectories and accumulation of floating marine litter in different meteo-oceanographic conditions, the Lagrangian transport model previously described was used to follow hypothetical floating marine litter originated from land- and sea-based sources, including wind drift and beaching effects. Figure 2 shows the results of the sea surface current obtained from model simulations on 9 January, 22 September and 10 October 2017, with the correspondent sea surface temperature (SST) retrieved on the European Meteorological Operational-B (MetOp-B) satellite image available.

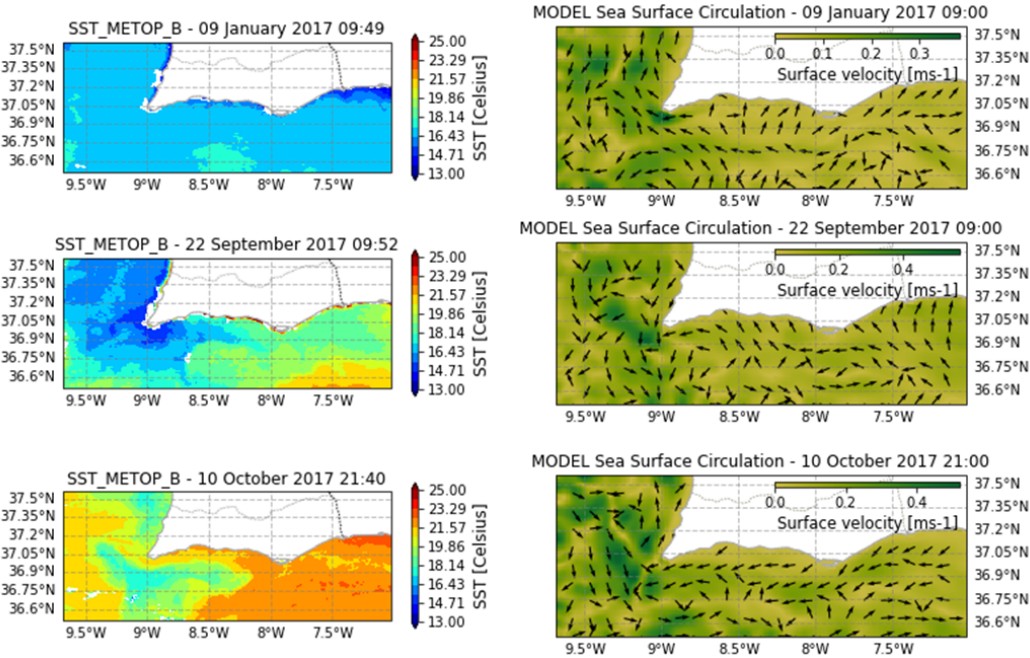

**Figure 2.** Snapshots of sea surface current field (m/s) predicted by the model on the right, while a snapshot of sea surface temperature (°C) obtained from satellite images on the left for 9 January (scenario 1), 22 September (scenario 2) and 10 October of 2017 (scenario 3).

The trajectories of particles from the used sources for each of the three runs, with and without additional wind drift, is shown in Figure 3. Black particles emitted offshore correspond to particles discharged by commercial vessels, whereas the ones emitted close to the shore correspond to particle discharge by fishing and aquaculture activities. Red dots are related to particles emitted by land-based sources, such as tourism and riverine.

Bigger dots represent the initial position of emission of the particles, whereas smaller dots are the mean trajectories calculated for each source in each simulated time-step.

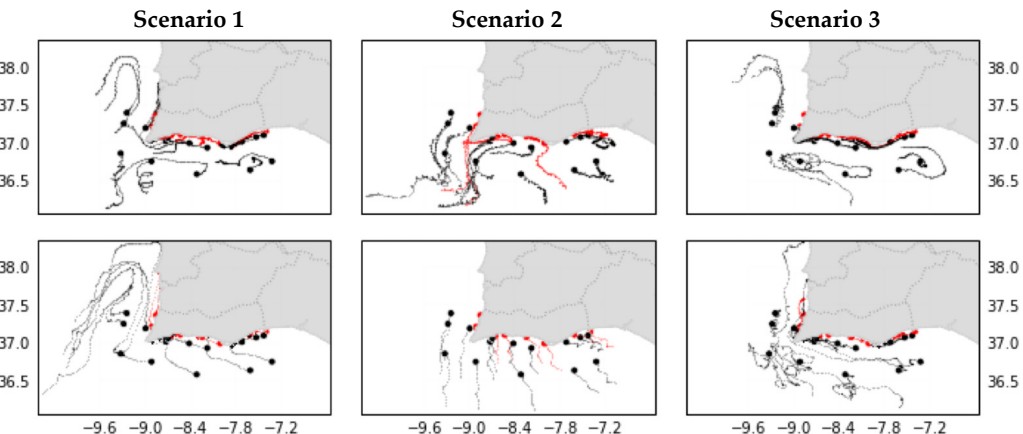

**Figure 3.** Trajectories of floating marine litter simulated for each classified period: without additional wind drift (**top**) and with 3% wind drift (**bottom**). The black dots and lines correspond to the initial discharge locations and trajectories of sea-based sources (fishing, aquaculture, and shipping), and the red dots and lines are associated with the discharge locations and trajectories of marine litter particles emitted by land-based sources (beaches and rivers).

Figure 4 displays the concentration of beached particles for each of the three runs, with and without the wind drift. In this study, the Algarve coast was sub-divided into 6 sub-regions for a better visualization of potential hot spots of marine litter on the coast. Those areas, identified in Figure 1, are: W-region between Aljezur and Praia da Bordeira, CSV-Cabo de São Vicente to Lagos, SWw-Lagos to Albufeira, SW-Albufeira to Faro, CSM-Faro to Tavira, and SE-Tavira to the Guadiana estuary. The concentration of particles is calculated by the number of particles per km of shoreline of each sub-region.

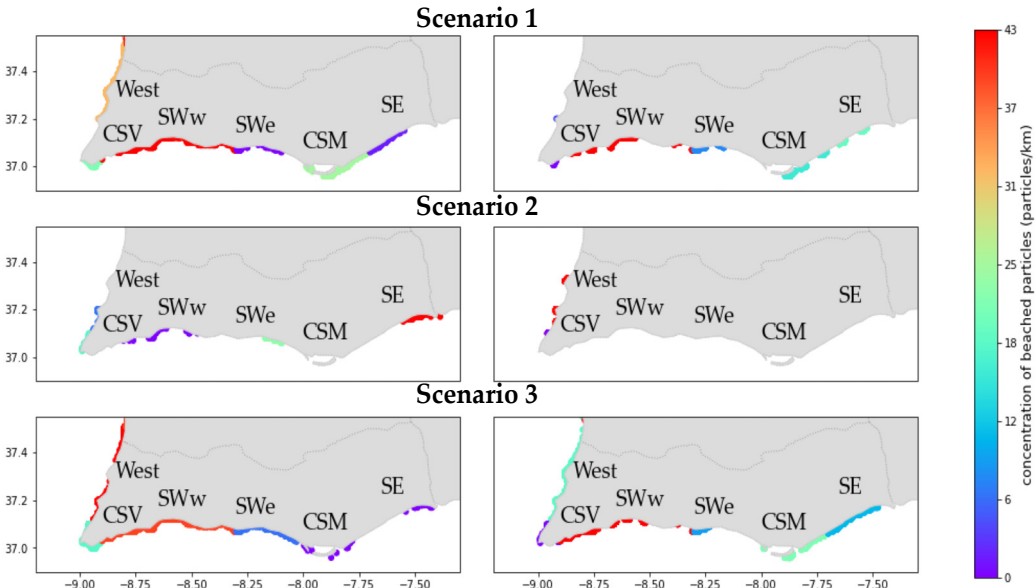

**Figure 4.** Concentration (particle/Km) of beached particles calculated for each of the 6 sub-regions and for each classified period: Scenario 1, Scenario 2, and Scenario 3 runs. Considering simulations without (**left**) and with the additional 3% wind drift (**right**).



*Monitoring Beach Litter*

Anthropogenic litter recorded in surveys performed by APA for the period of 2013 to 2017 were analysed here, to provide an understanding of the pollution in the Algarve shores. Therefore, results of item counts, types, and sources were displayed as aggregated results for both beaches surveyed in this study area: Faro Island and Praia da Batata. Table 2 displays the summary of data on composition and the sources of litter calculated by Litter Analyst for beaches in the south of Portugal.

**Table 2.** Item type and sources of litter found along the Algarve coast, with median and average per 100 m, percentage of total amount, trend (total items/year), and significance of trend for the period 2013–2017.

| Litter Classification | Litter Classification | Median | Average | % of Counts | Trend | Sig. of Trend (*p*-Value) |
|---|---|---|---|---|---|---|
| Material | Paper/Cardboard | 100.8 | 111.4 | 51.40% | 14.2 | 0.38 |
| | Plastic/Polystyrene | 66.8 | 80.9 | 37.30% | 1.8 | 0.80 |
| | Metal | 9.3 | 10.5 | 4.80% | 1.2 | 0.15 |
| | Wood | 4.3 | 4.8 | 2.20% | 0.1 | 0.53 |
| | Glass | 2.3 | 2.8 | 1.30% | 0.0 | 0.67 |
| | Sanitary | 1.8 | 2.5 | 1.20% | 0.5 | 0.05 |
| Source | Tourism | 121.5 | 144.9 | 66.80% | 16.7 | 0.22 |
| | Other | 51.8 | 53.7 | 24.80% | −2.6 | 0.27 |
| | Shipping | 8.0 | 10.7 | 4.90% | −0.5 | 0.70 |
| | Fishing | 2.5 | 4.7 | 2.20% | 0.2 | 0.65 |
| | Sanitation | 2.0 | 2.8 | 1.30% | 0.6 | 0.03 |

The analysis of individual litter types with the highest abundance on the shore, called Top-80% analysis, resulted in 10 items mostly observed with high abundance in both beaches monitored between 2013 to 2017. Table 3 displays the median, total abundance (% of counts), trend and trend significance ($p < 0.05$) of the first 5 items with highest abundance in the shore, wherein together sums more than half of all litter items surveyed in the surveys.

**Table 3.** The 5 first items in the Top-80% of most found items along the Algarve coast, with median and average count per 100 m, percentage of total amount, trend (counts/year), and significance of trend for the period 2013–2017.

| Sub-Region | Litter Type Definition | Median | Average | % of Counts | Trend | Sig. of Trend (*p*-Value) |
|---|---|---|---|---|---|---|
| South of Portugal (*n* = 23) | Cig_Stubs | 93.3 | 99.7 | 44.0% | 12.1 | 0.299 |
| | Plastic pol. pieces < 50 cm | 13.8 | 18.6 | 8.6% | −0.7 | 0.697 |
| | Nets and Ropes | 7.8 | 12.2 | 5.6% | −0.3 | 0.82 |
| | Plastic: Other | 7.0 | 8.2 | 3.2% | 0.9 | 0.434 |
| | Plastic: Small_bags | 6.5 | 7.9 | 3.6% | 0.2 | 0.845 |

Furthermore, the abundance of litter items observed during APA beach surveys in 2017 were used in this study with the aim of qualitatively validating the beaching results of all three simulations. Table 4 shows the total beached particles emitted by land and sea-based sources accumulated on two sub-regions, CSM and SWw, as well the total count of litter observed on both Faro Island and Praia da Batata surveyed by APA in winter, and autumn of 2017.

**Table 4.** Total counts of litter observed on beach litter surveys during 2017 from Faro Island and Batata beach, and total number of beached particles at CSM and SWw sub region modelling in this study, considering the three hydrodynamics and wind drift.

| Methods | Sources | Faro/CSM | | | Batata/SWw | | |
|---|---|---|---|---|---|---|---|
| | | Winter | Summer | Autunm | Winter | Summer | Autumn |
| Survey | Land | 144 | 288 | 371 | 70 | 72 | 246 |
| | Fishing | 17 | 4 | 8 | 20 | 0 | 8 |
| | Shipping | 39 | 74 | 69 | 35 | 17 | 33 |
| | Total | 200 | 366 | 448 | 125 | 89 | 287 |
| Model | Land | 544 | 0 | 12 | 1742 | 33 | 1252 |
| | Fishing | 32 | 0 | 0 | 0 | 0 | 152 |
| | Shipping | 0 | 0 | 0 | 0 | 0 | 0 |
| | Total | 576 | 0 | 12 | 1742 | 33 | 1404 |
| Model (3% Wd) | Land | 0 | 0 | 477 | 2461 | 0 | 2221 |
| | Fishing | 1000 | 0 | 0 | 2000 | 0 | 2889 |
| | Shipping | 1000 | 0 | 0 | 1000 | 0 | 0 |
| | Total | 2000 | 0 | 477 | 5461 | 0 | 5110 |

## 4. Discussion

The hydrodynamic results produced by SOMA reveal a good agreement between sea surface temperature (SST) features, presenting the major features observed from space and described in several previous studies [16,17,32,33]. An analysis of Figure 2 shows a similar pattern for the sea surface circulation in scenario 1 (1–15 January run) and 3 (7–22 October run). Both these scenarios are characterized by a westward and northward flow along the SW Iberian coast. However, in scenario 1, there is a brief episode of southerly wind and the temperature is lower near the coast compared to offshore. While, in scenario 3, easterly winds were predominant, and a warm counter current progressed near the Algarve coast. In contrast, in scenario 2, under prevailing northerly winds, a jet of cold upwelled the water flowing southward with a velocity of 0.2 m/s and took place on the southwestern coast, similar to what is described in [33]. However, the intrusion of west coast cold upwelled waters turning easterly in the south coast was not reproduced in the model (Figure 2). This difference between the model result and the SST satellite images in scenario 2 might be due to the factor of the input fields (winds and currents), which are provided by numerical models that may have their own errors.

Figures 3 and 4 showed that during scenarios 1 and 3, the transport and fate of marine litter had great similarity. For both scenarios 1 and 3, the results showed an alongshore westward transport of particles along the south Portuguese coast, whilst the northward flow on the west coast transport floating litter reaching Sines coastal waters in 5 days. In scenarios 1 and 3, the last days were followed by brief episodes of northerly winds which induced the offshore transport. Litter particles were washed ashore along the Algarve coastline in scenarios 1 and 3, with notable accumulation along the southwestern coastline (45 particles per km of coast). The southwestern region encompasses one of the main touristic regions (between Lagos and Albufeira), which receives 60% of the Algarve visitors, as well as the natural park of PNSACV (the Southwest Alentejo and Vicentine Coast Natural Park) located in Aljezur. Simultaneously, in scenario 1, an increase in stranded particles is observed in the Natural Park region (CSM sub-region).

In scenario 2, the coastal upwelling leads a southward flow of floating marine litter, and some marine litter was transported closer to commercial vessel lines (Figure 3), whereas some particles emitted on the southern coast were transported closer to the coast and responded to more local hydrodynamic forcing. Moreover, the coastal upwelling induces an offshore transport of surface waters, preventing floating marine litter from being washed

ashore, agreeing with [15] findings in the NW Iberian coast. For this study, under upwelling events, only 3% of the marine litter was washed ashore, concentrated mostly on the shores near the Guadiana estuary, as shown in Figure 3.

Comparing Figures 3 and 4 of floating marine litter with and without wind drift, it is possible to observe that small changes in the wind drift may have an important effect on the trajectories and accumulation of floating litter. The modelling results suggest that addition of wind drift can alter the trajectories and fate of floating marine litter Figures 3 and 4. Therefore, under the upwelling conditions in scenario 2, most of the floating litter travelled southward, leaving the main domain (Level 2 in Figure 1) after 5 days, whereas in scenario 1 and 3, most particles with wind drift travelled faster towards the coastline, and were consequently beached on shore near its sources. For example, in scenario 1, over 70% of marine litter with a wind drift were beached after three days compared to 14% of marine litter without a wind drift. Moreover, no significant differences could be observed in the location of the highest accumulation, except in scenario 2, where the highest accumulation was observed in the Natural Park in the west coast.

Furthermore, the wind drift coefficient was shown to have a large influence on the quantity of particle accumulation on the shores. In the simulation with wind drift (3%), the percent of particles ending up at the beach for each scenario was as follows: scenario 1: 78%; scenario 2: 5%; and scenario 3: 75%. These percentages are reduced significantly if wind drift is not applied (scenario 1: 26%, scenario 2: 2%; and scenario 3: 21%). This study reinforces the importance of wind drift coefficients in the movement of floating marine litter. Most studies that use wind drift use a coefficient between 0.01 to 0.06 [3,4,10]; nevertheless, an accurate value remains unknown. Although determining the wind drift coefficient of marine litter is challenging to specify and differ a lot among different buoyant items, research to define suitable coefficients should be a priority.

According to beach litter data collected during APA surveys between 2013–2017, the average litter on Algarve shorelines was around 186 items per 100 m of beach. There is an increasing trend in total litter counts (11.3) with no statistical significance ($p = 0.399$). Of the material groups, paper and plastic are the most dominant type of litter, which constitute 51% and 37% of the total items counted, respectively (Table 2). Although many studies classify plastic items as the most abundant litter, in this study the increase of the dominance of paper litter could be attributed to a high concentration of cigarette butts (44%) on the study shores (Table 3); the OSPAR surveys classify cigarette butts as paper and not plastic litter, contrary to many other studies. This problem is one of the reasons why the monitoring of marine litter should be carried out following a standardised approach, thus allowing researchers to compare their results with surveys in different regions.

Survey results showed that over 50% of total litter originates from tourism activities (for example, beachgoers littering directly on the coast), showing that land-based inputs are likely to be the key sources of marine anthropogenic litter in this region (Table 2). These results agree with those from previous analyses in other areas, though the proportions vary [34,35]. Moreover, results analysis has shown that litter originated from tourism activities, especially cigarette stubs, has an increased trend of 17 and 12 items per year, respectively, with no statistical significance ($p > 0.05$). Moreover, this study observed significant trends that were hampered by the large spatial variation in the abundance of beach waste between the two beaches in the region [36], and the substantial temporal variations of waste in unique locations of research.

Model results were qualitatively evaluated against litter data collected during APA field data surveys from 2017 (Table 2). The results showed a great presence of simulated beached litter particles on the southwestern coast in the scenario 3, which agreed with the measured litter in autumn at Praia da Batata; this survey was carried under a counter current event. This result strengthens the hypothesis that higher values of litter concentrations in the western region of the Algarve may be related to the predominant westward surface current. Moreover, both model and in situ data show high concentrations of beached

litter from land- and sea-based sources in the Praia da Batata beach and SWw sub-region during January.

Contrary to the monitoring data, the September simulation predicted zero presence of beached litter in the CSM sub-region (where Faro Island is located) under upwelling events during the summer period. Therefore, along the Algarve coast, the concentrations of litter on the beaches may also be explained by the increase in anthropogenic activities in this period. Faro beach attracts a high number of tourists. In 2017, for example, around 520.000 hotel nights were estimated in that area, with 36% of the hotel stays observed between July and September [37]. This could be the reason for the high values of the monitoring surveys on the Faro coast during summer, and not accounted for in the model. It is estimated that holidaymakers may cause a 40% surge in marine litter during summer [38].

## 5. Conclusions

In this study, a series of simulations were conducted to describe the fate and distribution of floating litter particles, after being released from specific source regions in the Algarve. Model analysis pictured that the meteo-oceanographic conditions and wind drift have a great influence on the distribution and fate of floating marine litter in the coast of the Algarve. In the period of upwelling, floating marine litter were transported offshore and travelled southward, while during the winter and counter current, litter particles were transported south-western and northward. A higher probability of marine litter accumulation along the coast occurred during the winter and counter current scenario, accumulating mainly along the southwestern coastline, between Albufeira to Aljezur. Beach litter monitoring and model results showed that land-based inputs, especially tourism, are likely key sources of marine anthropogenic litter, and therefore concentrations of litter on the Algarve shores may also be explained by the increase in beach visitors during the summer. The beach areas of high floating litter concentrations may be considered as a priority for future surveys; this will help to assess the status of floating litter pollution in the Algarve coast. For future studies, it is crucial to increase the monitoring of marine litter in the sea surface and beaches following the standard approach to improve the credibility of future model studies in the study area.

**Author Contributions:** F.M. and J.J. developed the study concept and E.R. performed the simulations and analysis under the supervision of F.M. and J.J. The manuscript was written by E.R. and reviewed and edited by all the other authors. F.M. was responsible for funding. All authors have read and agreed to the published version of the manuscript.

**Funding:** This research was funded by OCASO, grant number Interreg POCTEP project GA 0223_OCASO_5_E, and Interreg POCTEP project GA 0754_CIU3A_5_E.

**Institutional Review Board Statement:** Not applicable.

**Informed Consent Statement:** Not applicable.

**Acknowledgments:** The authors special thanks to Directorate-General for Natural Resources, Safety and Marine Service (DGRM) and Portuguese Environment Agency (APA) for supplying in-situ litter data collected during the monitoring of Algarve beaches. Special thanks to Lara Mills for reviewing the grammatical mistakes in this article. Thanks to the Centre of Marine and Environment research in the Ualg and Nautilos project, which has received funding from the European Union's Horizon 2020 research and innovation programme under grant agreement No. 101000825. Thanks also to the three reviewers for their helpful and constructive comments on the manuscript.

**Conflicts of Interest:** The authors declare no conflict of interest.

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
