# Peer review of "Marine Litter on the Coast of the Algarve: Main Sources and Distribution Using a Modeling Approach"

_jmse, doi:10.3390/jmse9040412_

Round 1

Reviewer 1 Report

In line 24, and later in the text, it is not clear how tourism generates marine litter. There is a need to include a paragraph somewhere to explain how it does generate.

Line 50-51 It is not clear what demographic means, explain this.

Line 119 downscale - add an s to generate "downscales"

It is not clear what the word Lagrangian means, and this needs to be explained.

Lines 223 & 225. Use capitals A and B not (a) and (b).

In lines 330 and 331 it is not clear why (    ) is presented. This is also the case in lines 341 and 342 and 344 and 345 and 351 and 352.

However, the paper is quite interesting and mostly well written

Author Response

Dear Dr./ Mr./Ms. Editor,

Thank you for allowing me to submit a revised draft of my manuscript titled Marine Litter on the Coast of the Algarve: Main Sources and Distribution Using a Modeling Approach to JMSE-Marine Litter special issue. We appreciate the time and effort that you reviewers have dedicated to providing your valuable feedback on my manuscript. We are thankful to the reviewers for their insightful comments on my paper. We have been able to incorporate changes to reflect most of the suggestions provided by the reviewers. And changes were highlighted within the manuscript.

Comments from Reviewer 1

Comment 1: In line 24, and later in the text, it is not clear how tourism generates marine litter. There is a need to include a paragraph somewhere to explain how it does generate.

Response: Thank you for pointing this out. We agree with this comment. Therefore, tourism activities were replaced by holidaymakers and beachgoers which can cause littering directly on the shoreline.

Comment 2: It is not clear what the word Lagrangian means, and this needs to be explained.

Response: We agree that that the explanation of the Lagrangian approach can assists readers less familiar with modelling terminology. The brief explanation added was; the Lagrangian framework describes the motion of the virtual particle in the flow field in the function of time, for and paper with an extensive explanation of the approach were add in the text as well.

Regarding the loose parenthesis in the discussion; it appears to me that an error occurred when the manuscript was converted to the magazine format, and the text inside of the parenthesis were deleted. However, the text which introduces the table in the paragraph (e.g. Table 1) were added again to the manuscript.

In addition to the above comments, all spelling and grammatical errors pointed out by the reviewers have been corrected and revised by a native English speaker.

Reviewer 2 Report

The manuscript presents a quantitative assessment of litter sources and movement in the coastal region of the Algarve, the southernmost province of Portugal. The authors introduce their subject in reasonable fashion - with a potentially excessive superlative, e.g., "critical problem". The general problem of ocean pollution can be considered critical, but the authors have not shown that it has crisis dimensions on the Portuguese coast. 

The problems noted are 1) Figure 2 is too small , and 2) the text abounds in grammatical errors and unclear or awkward passages. Examples from pages 1 and 2 follow. 

p. 1 line 16 "Model" should be "the model", or "Modelling suggests . . . "

p. 1 line 33. "reduces the tourism" should be "reduces tourism", and "municipalities are forced to invest in beach-cleaning strategies" should be "municipalities are forced to clean beaches".

p. 1 line 39 is ungrammatical

p. 2 lines 55-56 "the distribution of litter along the Portuguese coast remains scarce". Presumably, the authors mean that "information on the distribution of litter remains scarce"

p. 2 line 61. "met-ocean conditions" 

p 2 line 94  "...  presence of litter . .. on diets of aquatic birds" should be "in the diets of aquatic birds"; better "in the stomachs of aquatic birds" ? 

The authors should revise the paper with the help of a professional editor or native speaker.

A more complex issue is the use of specialized terms relating to physical modeling: e.g. "object-oriented modeling", "primitive Navier-Stokes equations", "discretization", "z-level layers", and clarifying the difference between hydrodynamic and Lagrangian modeling. While these usages are appropriate, it would aid readers less familiar with modeling terminology if more definitions and clarifications could be incorporated in the text. 

Author Response

Dear Dr./ Mr./Ms. Editor,

Thank you for allowing me to submit a revised draft of my manuscript titled Marine Litter on the Coast of the Algarve: Main Sources and Distribution Using a Modeling Approach to JMSE-Marine Litter special issue. We appreciate the time and effort that you reviewers have dedicated to providing your valuable feedback on my manuscript. We are thankful to the reviewers for their insightful comments on my paper. We have been able to incorporate changes to reflect most of the suggestions provided by the reviewers. And changes were highlighted within the manuscript.

Comments from Reviewer 2:

Comment 1:The problems noted are 1) Figure 2 is too small

Response: Thank you for pointing this out. We agree with this comment, so figure 2 and the text were improved. In figures 2 and 4 the font size inside in both figures was increased, the arrows in Figure 2 were also increased. It can be observed in line 244 and 268 of the manuscript.

Comment 2:grammatical errors and unclear or awkward passages in line 39.

Response: The paragraph about the marine litter in MSFD was modified, clarifying the awkward passage and correcting the grammatical errors. The new paragraph is:

“ In Europe, marine litter is recognized by the Marine Strategy Framework Directive (MSFD). The MSFD is a framework that requires the members of the state to develop and apply programs of measures to achieve or maintain Good Environmental Status (GES) in European Seas. Moreover, the MSFD GES technical subgroup on marine litter identified various priority research for a better assessment of marine litter, including; evaluate the factors that affect the transport and fate, development of standardization of monitoring methodologies, and use of modelling to assist in monitoring and management.”

Comment 4:  grammatical errors in line 16,33, 55-56, 61 and 94

Response: we agreed with the suggestions and the words were corrected.

Comment 3:A more complex issue is the use of specialized terms relating to physical modeling: e.g. "object-oriented modeling", "primitive Navier-Stokes equations", "discretization", "z-level layers", and clarifying the difference between hydrodynamic and Lagrangian modeling.

Response:A brief explanation about the object-oriented modelling and primitive Navier-stokes equations were added to the figure as a footnote on page 4, to and help clarify readers that are not familiar with the modelling terminology. Moreover, discretization was replaced by vertical resolution and z-level layers for vertical layers. 

  • Object-Oriented programming isolates different parts of the code and allows communication among them using simple and robust interfaces.
  • The Navier-Stokes equations describe the motion of fluids and are the fundamental equations of fluid dynamics. In ocean modelling, approximations are used to solve these equations, written in the coordinate frame of reference fixed to the rotating earth, which, together with an equation of state (relating pressure, temperature and salinity) and equations for turbulent quantities, allow us to compute the ocean velocity fields.

Comment 4: Concerning the methods and results wherein the option Must be improved were selected.

Response: To improve these two sections of the manuscript, some modifications were made;

  1. Brief explanation of the Lagrangian approach was added, as well as the authors agreed the Lagrangian module was not well explained and should be improved. Therefore, a new and more detailed explanation of the module was added:

“To simulate the floating marine litter trajectories, the Lagrangian module based on MOHID Water system is used. The Lagrangian framework describes the motion of virtual particles in the velocity field as a function of time [25]. The spatial transport of the particles is the sum of four velocities:

Equations 1 and 2, in lines 134-138

“Looking at Equation 2, u1i is the is the surface current velocity from the hydrodynamic module. In this study, both hydrodynamic and Lagrangian modules run simultaneously and there is an exchange of turbulence information in real time between both models, thus increasing the accuracy of the computed trajectories [22]. In addition, the multi-mesh approach ensures that the high-resolution hydrodynamics (Level 2) is selected whenever the particles move into their geographical boundaries.  u2i  is the wind velocity at a height of 10 m over the sea surface, and Cis the user-defined wind drag coefficient. Wind drag factor mostly compensates for the lack of wind stress in the upper few centimeters of hydrodynamic models to drive the marine litter immersed at the water. This approach is applied in hydrodynamic models with the vertical thickness of the top layer not sufficiently small. This situation would require a resolution at the surface of the order of a few centimeters, which is currently not computationally feasible. This drag velocity is also useful to account for the effect of the wind in the exposed part of emerged litter. Many studies reported that the wind induced speed of floating marine litter at the sea surface, and values of wind drag coefficient normally varies from 1% to 5% [3-4,10].  u3i the random velocity due diffusion transport, and finally u4i  is the stokes drifts. Although stokes drift has been reported as an important mechanism that affects the trajectories of floating marine litter, in this work it was not considered.”

Response :In the result section, a short introductory context of the manuscript was added to help the readers to understand the results, and the text reporting the findings of hydrodynamic in Figure 2 was modified;

“3. Results

To assess the trajectories and accumulation of floating marine litter in different meteo-oceanographic condition, the Lagrangian transport model previously described was used to follow hypothetical floating marine litter originated from land and sea-based sources, including wind drift and beaching effects. Figure 2 shows the results of the sea surface current obtained from model simulations on January 9th, September 22nd and October 10th of 2017, with the correspondent sea surface temperature (SST) retrieved on the European Meteorological Operational-B (MetOp-B) satellite image available.  “

In addition to the above comments, all spelling and grammatical errors pointed out by the reviewers have been corrected and revised by an English native speaker.
